# The MEK/ERK/miR-21 Signaling Is Critical in Osimertinib Resistance in EGFR-Mutant Non-Small Cell Lung Cancer Cells

**DOI:** 10.3390/cancers13236005

**Published:** 2021-11-29

**Authors:** Wen-Chien Huang, Vijesh Kumar Yadav, Wei-Hong Cheng, Chun-Hua Wang, Ming-Shou Hsieh, Ting-Yi Huang, Shiou-Fu Lin, Chi-Tai Yeh, Kuang-Tai Kuo

**Affiliations:** 1Department of Medicine, MacKay Medical College, New Taipei City 252, Taiwan; 20236@s.tmu.edu.tw; 2Division of Thoracic Surgery, Department of Surgery, MacKay Memorial Hospital, Taipei 104, Taiwan; 3Department of Medical Research & Education, Taipei Medical University—Shuang Ho Hospital, New Taipei City 235, Taiwan; 20604@s.tmu.edu.tw (V.K.Y.); 20213@s.tmu.edu.tw (M.-S.H.); 15729@s.tmu.edu.tw (T.-Y.H.); ctyeh@s.tmu.edu.tw (C.-T.Y.); 4Division of Hematology and Oncology, Department of Internal Medicine, Taipei Medical University—Shuang Ho Hospital, New Taipei City 235, Taiwan; 13520@s.tmu.edu.tw; 5Department of Dermatology, Taipei Tzu Chi Hospital, Buddhist Tzu Chi Medical Foundation, New Taipei City, 231, Taiwan; 10205@s.tmu.edu.tw; 6School of Medicine, Buddhist Tzu Chi University, Hualien 970, Taiwan; 7Department of Pathology, Taipei Medical University—Shuang Ho Hospital, New Taipei City 235, Taiwan; 18011@s.tmu.edu.tw; 8Department of Medical Laboratory Science and Biotechnology, Yuanpei University of Medical Technology, Hsinchu 300, Taiwan; 9Division of Thoracic Surgery, Department of Surgery, School of Medicine, College of Medicine, Taipei Medical University, Taipei 110, Taiwan; 10Division of Thoracic Surgery, Department of Surgery, Taipei Medical University—Shuang Ho Hospital, New Taipei City 235, Taiwan

**Keywords:** non-small cell lung cancer (NSCLC), Osimertinib resistance, MEK inhibitor Trametinib, miR-21, cancer-associated fibroblasts (CAFs), human lung fibroblasts (HLFs)

## Abstract

**Simple Summary:**

Our study provided data that the inhibition of MEK/ERK signaling could overcome Osimertinib resistance both in vitro and in vivo. Mechanistically, MEK inhibitor Trametinib suppressed the tumorigenic properties of NSCLC cells by reducing the generation of CAFs. The trametinib-mediated anti-cancer function was also associated with the significantly suppressed level of miR-21, of which primary targets included PDCD4, as shown in this study and MEK inhibitor Trametinib significantly suppressed Osimertinib-resistant NSCLC tumor growth by abolishing both processes.

**Abstract:**

Background: The third-generation epidermal growth factor receptor (EGFR) inhibitor, Osimertinib, is used to treat non-small cell lung cancer (NSCLC) patients with tyrosine kinase inhibitor (TKI) resistance caused by acquired EGFR T790M mutation. However, patients eventually develop resistance against Osimertinib with mechanisms not yet fully clarified. Activated alternative survival pathways within the tumor cells and cancer-associated fibroblasts (CAFs) have been proposed to contribute to Osimertinib resistance. MET and MEK inhibitors may overcome EGFR-independent resistance. Another acquired resistance mechanism of EGFR-TKI is the up-regulation of the RAS/RAF/MEK/ERK signaling pathway, which is the key to cell survival and proliferation; this may occur downstream of various other signaling pathways. In this report, we reveal the possible regulatory mechanism and inhibitory effect of the MEK inhibitor trametinib applied to MEK/ERK/miR-21 axis and PDCD4 in Osimertinib resistance. We found a possible regulatory role of PDCD4 in ERK signaling. PDCD4 is a new type of tumor suppressor that has multiple functions of inhibiting cell growth, tumor invasion, metastasis, and inducing apoptosis. Previous bioinformatics analysis has confirmed that PDCD4 contains the binding site of miR-21 and acts as a tumor suppressor in the regulation of various processes associated with the development of cancer, including cell proliferation, invasion, metastasis, and neoplastic transformation. Based on the above analysis, we hypothesized that the tumor suppressor PDCD4 is one of the effective inhibitory targets of miR-21-5p. Methods: The expression between EGFR and ERK2 in lung adenocarcinoma was evaluated from the TCGA database. Osimertinib-sensitive and resistant NSCLC cells obtained from patients were used to co-culture with human lung fibroblasts (HLFs) to generate CAF cells (termed CAF_R1 and CAF_S1), and the functional roles of these CAF cells plus the regulatory mechanisms were further explored. Then, MEK inhibitor Trametinib with or without Osimertinib was applied in xenograft model derived from patients to validate the effects on growth inhibition of Osimertinib-resistant NSCLC tumors. Result: ERK2 expression correlated with EGFR expression and higher ERK2 level was associated with worse prognosis of patients and Osimertinib resistance. CAFs derived from Osimertinib-resistant cells secreted more IL-6, IL-8, and hepatocyte growth factor (HGF), expressed stronger CAF markers including α-smooth muscle actin (α-SMA), fibroblast activation protein (FAP) plus platelet-derived growth factor receptor (PDGFR), and enhanced stemness and Osimertinib resistance in NSCLC cells. Meanwhile, increased MEK/ERK/miR-21 expressions were found in both CAFs and NSCLC cells. MEK inhibitor Trametinib significantly abrogated the abovementioned effects by modulating β-catenin, STAT3, and ERK. The xenograft model showed combining Osimertinib and Trametinib resulted in the most prominent growth inhibition of Osimertinib-resistant NSCLC tumors. Conclusions: Our results suggested that MEK/ERK/miR-21 signaling is critical in Osimertinib resistance and CAF transformation of NSCLC cells, and MEK inhibitor Trametinib significantly suppressed Osimertinib-resistant NSCLC tumor growth by abolishing both processes.

## 1. Introduction

Lung cancer is the leading cause of cancer death worldwide, with an estimated global mortality of nearly 1.76 million in 2018 [1]. Dysfunctional and activating epidermal growth factor receptor (EGFR) is one of the major oncogenic drivers in non-small cell lung cancer (NSCLC). Over the most recent decade, the discovery of targeted therapeutic agents, EGFR tyrosine kinase inhibitors (TKIs), has revolutionized the treatment of lung cancer. Patients with lung adenocarcinomas carrying EGFR mutations (mainly exon-19 deletion and L858R) showed positive responses towards TKIs such as Gefitinib and Afatinib, the first and second generations of TKIs. Currently, most credible clinical guidelines recommend EGFR TKIs as the first-line treatment for EGFR-mutant metastatic NSCLC patients [2]. Unfortunately, most of the initial TKI responders inevitably develop resistance and experience disease relapse [3,4,5]. Several mechanisms of acquired resistance to EGFR TKIs have been reported [6]. Among them, the emergence of EGFR T790M mutation has been well-established and contributes to more than 50% of all cases [7,8,9]. This spurred the development of the third-generation EGFR TKI, namely Osimertinib (AZD9291). It irreversibly targets the ATP-binding site C797 of EGFR tyrosine kinase and shows activity against diverse EGFR mutations with or without T790M, resulting in impressive responses in EGFR T790M-positive patients [10,11]. 

Nevertheless, similar to previous generations of TKIs, NSCLC patients eventually develop resistance against Osimertinib, via EGFR-dependent and EGFR-independent pathways. The EGFR-dependent pathway includes the acquisition of C797S mutation [12,13], as well as other less frequent mutations such as G796 and L792 [14,15], on the kinase-binding site of EGFR. The EGFR-independent pathway includes activation of other receptor tyrosine kinases (RTKs) which share the common downstream pathways of EGFR, like HER2 and MET [16,17], or working via other mutated genes, for example, RAS and BRAF V600E [18,19]. Based on these clinical observations, it appears that TKI treatment inevitably drives the tumor cells to adapt to the treatment and acquire resistance. Thus, the development of alternative strategies which take into account multiple targets should be considered.

The tumor microenvironment (TME) has been well recognized as an essential component for tumorigenesis. One of the major players within the tumor microenvironment is the cancer-associated fibroblasts (CAFs). CAFs are one of the most abundant components in the TME and have the ability to promote tumor progression. CAFs promote angiogenesis and migration by synthesizing favorable extracellular matrix (ECM) [20,21] and help immune evasion by recruiting immunosuppressive cells via the production of growth factors and inflammatory cytokines [22,23]. Meanwhile, HGF (hepatocyte growth factor) produced by CAFs, in turn, activates the MAPK and PI3K/AKT pathways in the tumor cells, leading to resistance to the treatments [24,25]. Collectively, tumor cells and CAFs have intimate crosstalk to promote cancer progression, and targeting tumor cells alone may be insufficient. Therefore, the elimination of CAFs should also be considered when constructing therapeutic interventions. Meanwhile, with the growing evidence of the role of small non-coding RNA, particularly micro-RNA (miRNA), in altering and regulating many key biological processes of various cancers, the important role of miR-21 in lung cancer has also been addressed [26].

In this study, we first used the TCGA lung adenocarcinoma database to recognize the correlation between EGFR and ERK expression and evaluated the association between ERK and Osimertinib resistance in tissue pairs. Then we induced CAFs by co-culturing NSCLC cells deriving from patients of different Osimertinib sensitivity with human lung fibroblasts (HLF) and investigated the characteristics CAFs and NSCLC cells. The role of the MEK/ERK/miR-21 pathway was further investigated. We also applied Trametinib (a MEK1/2 inhibitor) to verify the suppressive effects on tumorigenesis of NSCLCs and CAFs. Finally, using a patient-derived xenograft model, we demonstrated the efficacy of Trametinib in combination with Osimertinib to inhibit the growth of Osimertinib-resistant cancer cells *in vivo*. 

## 2. Materials and Methods

### 2.1. Cell Culture and Clinical Sample Acquisition

Human lung cancer cell lines, H1975 (with activating EGFR L858R+T790M mutations) and normal human lung fibroblasts (HLF, PCS-201-013) were purchased from ATCC (American Type Culture Collection, Manassas, VA, USA) and maintained according to the conditions suggested by ATCC. Clinical samples from patients diagnosed with non-small cell lung cancer (NSCLC) resistance and responding to Osimertinib treatment were procured from Taipei Medical University-Shuang Ho Hospital (Approval no.: JIRB N201801066). All patients were explained under the Declaration of Helsinki and given informed consent in writing before procedures. The clinicopathological features are listed in Appendix A. NSCLC cells resistant and sensitive to Osimertinib were cultured from a series of tumor specimens. The representative examples of Osimertinib-resistant and Osimertinib-sensitive cell lines deriving from patients are shown in Appendix A. For tumor primary cell culture, NSCLC tumor tissues were washed in sterile 1× phosphate-buffered saline (PBS) containing penicillin and streptomycin. Specimens were finely minced and placed in a serum-free RPMI medium. A single-cell suspension was obtained by performing a series of mechanical dissociations and enzymatic disaggregation (Trypsin and collagenase) followed by filtration through a 40-μm filter. Red blood cells were removed by hypotonic lysis. Finally, cells were resuspended in serum-free medium, counted using trypan blue to exclude dead cells, and plated at a density of 2 × 10^4^ viable cells per ml. All Osimertinib resistant clones were maintained RPMI complete medium supplemented with 1.5 µM Osimertinib (3 months). Samples resistant to Osimertinib (termed P_AZDR1) and sensitive to Osimertinib (termed P_AZDS1) were collected, maintained, and expanded for further experiments. The experimental process is shown in Appendix A. Patients were fully informed, and a written consent form was signed before the operation. The pathological examination was performed by the Department of Pathology in Taipei Medical University-Shuang Ho Hospital and all verified cases met the criteria of NSCLC. 

### 2.2. Cell Viability Test

Cell viability was tested using Cell Counter Kit-8 (CCK-8, Kumamoto, Japan). NSCLC cells were seeded (with a density of 5.0 × 10^3^ cells per well) in 96-well plates and treated with Osimertinib (AZD9291, Catalog No. S7297, SelleckChem, Houston, TX, USA) and/or Trametinib (GSK1120212, Catalog No. S2673, SelleckChem, Houston, TX, USA) at indicated concentrations (0 to 30 µM for 48 h) before the CCK-8 assay was performed. The optical density readings were determined at 450 nm. Cell viability was then expressed in percentage.

### 2.3. The NSCLC and HLF Cell Co-Culture Experiments

NSCLC cells and HLF cells were co-cultured using a Boyden chamber system, where cancer cells were seeded in the upper chamber (1 × 10^5^ cells) and HLF at the bottom (1 × 10^4^ cells) without refreshing culture media, for 4 days. The CAF cells were generated from co-culturing HLFs with P_AZDR1 cells (termed CAF_R1) and P_AZDS1 cells (termed CAF_S1). Resultant CAF cells were harvested for further analyses. For Osimertinib resistance assays, NSCLC cells were first co-cultured with CAF cells for 72 h, before the addition of Trametinib (0–30 µM, 48 h). H1975+CAF_R1 cells were established by long-term (approximately 50 passages) co-culturing with CAF cells generated from P_AZDR1 experiments. 

### 2.4. Quantitative PCR (qPCR) Analysis

Cellular total RNA was extracted using TRIzol reagent (Invitrogen, Waltham, MA, USA). SYBR Premix Ex TaqII kit (Takara Bio, Taipei, Taiwan) was utilized to perform qPCR experiments with 7500 Fast Real-Time PCR System (Applied Biosystems, Thermo Fisher Scientific). GAPDH served as the housekeeping gene for mRNAs and U6 for miR-21 experiments. The relative expression was determined by the comparative qPCR method. Primer sequences are listed in Appendix A. 

### 2.5. Sphere Formation Assay 

Serum-deprived culture conditions were used to grow tumor spheres to examine the self-renewal ability of the NSCLC cells in response to different treatments, sphere formation assay was performed as described previously [27]. NSCLC cells (5 × 10^3^ per well) were plated in ultra-low-attachment six-well plates (Corning). NSCLC cells were able to form floating spheres after incubation for 7–14 days, the spheres formed were counted and photographed.

### 2.6. SDS-PAGE and Western Blots

Protein lysates from NSCLC cells were obtained using RIPA buffer (Millipore) with protease/phosphatase inhibitor mixture (1:100) and PMSF (Beyotime). Protein quantification was performed by a BCA Protein Assay Kit (Thermo Fisher Scientific, Rockford, IL, USA). Samples were then separated by 10% SDS-PAGE and transferred to PVDF membranes (0.45, Millipore). The membrane was blocked in TBST (with 5% non-fat milk) washed and incubated with primary antibodies (4 °C, overnight). MEK1/2 (1:500, Abcam, ab178876), ERK1/2 (1:500, Abcam, ab184699), STAT3 (1:800, Abcam, ab68153), β-catenin (1:1000, Abcam, ab68183), PDCD4 (1:1000, Abcam, ab79405), β-actin (1:1000, Abcam, ab115777). Immuno-detection was obtained using a chemiluminescence kit (ECL; Thermo Fisher Scientific) and images were captured and analyzed using the UVP BioDoc-It system (Upland, CA, USA). 

### 2.7. MicroRNA, miR-21 Transfection

For modulating miR-21-5p level in vitro to demonstrate its function in lung tumorigenesis. NSCLC cancer cells (cell lines or clinical samples) were cultured and maintained as described in previous sections. Commercially available, sequence validated exogenous miR-21-5p, inhibitor (oligonucleotides), and mimic molecules were purchased for this study. Mimic and inhibitor molecules were transfected into NSCLC cells using Lipofectamine 2000 (Invitrogen, Carlsbad, CA, USA) according to the protocol provided by the manufacturer (Academia Sinica, Taipei, Taiwan). Transfected cells were maintained as described in the cell culture section. The transfected cells were maintained in DMEM medium (GIBCO, Carlsbad, CA, USA) supplemented with 10% fetal bovine serum (GIBCO) in a 5% CO_2_ humidified incubator at 37 °C. The cells were washed three times in PBS (pH 7.4) to remove the culture medium and serum before transfection.

### 2.8. Preclinical Evaluation of the Efficacy of Trametinib and Osimertinib Combination 

We established a patient-derived xenograft (PDX) mouse model by injecting P_AZDR1 cells (mixed with Matrigel, 7 × 10^6^ cells) subcutaneously. Twenty-four 8-week-old female nonobese diabetic (NOD)/severe combined immunodeficient (SCID) mice obtained from BioLASCO Taiwan Co., Ltd. (Taipei, Taiwan) were bred under standard experimental pathogen-free conditions. The in vivo studies were approved by the Institutional Animal Care and Use Committee (IACUC) of the Taipei Medical University (LAC-2017-0503). NSG mice were first anesthetized with xylazine (i.p, 8 mg/kg) and ketamine (120 mg/kg). Mice were randomly divided into 4 groups (six per group) and received different treatments, vehicle group (PBS P.O, 5 times/week), Trametinib alone (10 mg/kg, P.O, 5 times/week), Osimertinib alone (5 mg/kg, P.O, 5 times/week) and combination of both drugs (combined both regimens). The tumor volume was measured by standard caliper every other week using the following formula: V = (L × W2)/2 where L = long axis; W = width. Animals were humanely sacrificed following experiments, and their tumor and tissue samples were collected for further analyses.

### 2.9. Immunostaining of Tumor Sections

Tumor samples were harvested and fixed using formalin, followed by paraffin-embedment. Sections were prepared into sections (5-µm thickness) for staining protocol. Sections were first dewaxed in xylene and rehydrated with serial ethanol gradients and washed in deionized water. Subsequently, sections were blocked in PBS with 1% BSA, 1% donkey serum and 0.3% Triton X-100 and 0.01% sodium azide for 1 h, room temperature. After blocking, the slides were exposed to MEK (1:200, Abcam, ab32576), STAT3 (1:200, Abcam, ab68153), PDCD4 (1:200, Abcam, ab80590), and FAP antibody (1:200, Abcam, ab227703) at 4 °C overnight, washed and incubated in biotinylated link universal antiserum for 1 h at room temperature. Primary antibodies were diluted in blocking buffer and incubated in the cold overnight. The next day, sections were washed in PBST 3 times and incubated with secondary antibodies in blocking buffer (1 h, room temperature). Stained sections were then observed under a microscope and micrographed using a mounted digital camera.

### 2.10. Statistical Analysis

Experimental data were expressed as the mean + SD (standard deviation) and depicted using GraphPad Prism software (GraphPad Software, Inc., San Diego, CA, USA). The Student-T test was used for determining the difference between the two groups. Comparisons among several groups were analyzed based on one-way ANOVAs with Tukey’s post-hoc tests. *p* < 0.05 is recognized as statistically significant. Survival analysis was estimated using the Kaplan–Meier method along with the log-rank test to calculate differences between the curves. All experiments were repeated at least three times.

## 3. Results

### 3.1. Increased ERK Expression Is Found in Osimertinib-Resistant Cells and Associated with Poor Prognosis of NSCLC Patients

The TCGA lung cancer database analysis (N = 514) revealed that both EGFR/ERK2 (MAPK1) expression is significantly increased in patients with lung cancer, with a correlation coefficient value of 0.2508 (Figure 1A). Comparative gene analysis consisting of normal, fetal lung tissues and lung adenocarcinoma cells showed a significantly higher level of ERK2 (MAPK1) in the adenocarcinoma cells (Figure 1B). ERK2 (MAPK1) was found to be in the top 3% of genes elevated in the cohort and 2.4-fold higher than normal lung tissue. Using real-time PCR analysis, we found clinical lung cancer samples of Osimertinib-resistant tumors had higher levels of ERK2 (MAPK1) mRNA, as compared to Osimertinib-sensitive counterparts (Figure 1C). Elevated ERK2 (MAPK1) expression was significantly associated with poor survival in patients with NSCLC, from the cohort including 1925 patients (Figure 1D).

### 3.2. Osimertinib-Resistant NSCLC Cells Promoted CAF Generation with an Increased Level of miR-21 and MEK/ERK Expression

Cancer-associated fibroblasts (CAFs) are major contributors of malignant properties including drug resistance. Here, we co-cultured clinically obtained Osimertinib-resistant and Osimertinib-sensitive NSCLC cells (termed P_AZDR1, P_AZDS1, respectively) with normal human lung fibroblasts (HLF, PCS-201-013, ATCC) to obtain different CAFs, which were named CAF_R1 cells by mixing HLFs with P_AZDR1 cells and CAF_S1 cells by mixing HLFs with P_AZDS1 cells. Real-time PCR analysis results showed that CAF_R1 cells contained significantly higher CAF markers including increased FAP (fibroblast activation protein), α-smooth muscle actin (α-SMA), VIM (vimentin), and PDGFR (platelet-derived growth factor receptor) (Figure 2A). Comparatively, CAF_R1 cells induced a higher level of CAF markers than CAF_S1 cells. Since, miR-21 has been shown to be overexpressed in CAFs, lung tumor cells and involved in disease progression, we also evaluated miR-21 levels in different fibroblast cells. The miR-21 levels were highest in CAF_R1, followed by CAF_S1 and HLFs (Figure 2B). In addition, CAF_R1 cells secreted substantially more inflammatory cytokines including IL-6, IL-8, and HGF (hepatocyte growth factor, Figure 2C). More importantly, CAF_R1 cells facilitated the development of Osimertinib resistance in P_AZDS1 and H1975 cells (Figure 2D). Meanwhile, CAF_R1 cells also stimulated H1975 and P_AZDS1 cells to enhance the ability of tumor sphere formation (Figure 2E). The phenotype of increased tumorigenesis by CAF_R1 cells was documented to be associated with increased expression of MEK1/2, ERK1/2, STAT3, and β-catenin expression (Figure 2F).

### 3.3. Trametinib Significantly Reduced Tumorigenic Properties of Osimertinib-Resistant NSCLC Cells

In order to confirm the change in the amount of CAF when Trametinib is co-cultured, we evaluated the potential therapeutic effects of MEK inhibitor Trametinib for overcoming Osimertinib resistance using P_AZDR1 and H1975+CAF_R1 cells (co-cultured with CAF_R1). Assays for cell viability may monitor the number of cells over time, the number of cellular divisions, metabolic activity, or DNA synthesis. We first observed that Trametinib treatment (0.1 μM, 24 h) followed by Osimertinib (2 μM, 48 h) significantly reduced the cell viability of both P-AZDR1 and H1975+CAF_R1 cells (Figure 3A) with the respective IC50 values of Osimertinib as 0.5 µM and 0.25 µM, respectively. Equally important, a low concentration of Trametinib (0.1 μM) significantly reduced the mRNA level of FAP, VIM, α-SMA, and PDGFR in CAF_R1 cells (Figure 3B), indicative of reduced CAF phenotypic presentations. Consistently, the miR-21 level was significantly lowered in Trametinib-treated P_AZDR1 and H1975+CAF_R1 cells (Figure 3C). CAF-associated cytokines, including IL-6, IL-8, and HGF, were also reduced in the culture medium after Trametinib treatment (0.1 µM, 24 h, Figure 3D). Notably, Trametinib-treated P_AZDR1 and H1975+CAF_R1 cells were significantly less capable of generating tumor spheres as compared with the non-treatment counterparts (Figure 3E). Western blots of lysates from Trametinib treated P_AZDR1 and H1975+CAF_R1 cells showed a prominently suppressed level of MEK1/2, ERK1/2, STAT3, and β-catenin (Figure 3F).

### 3.4. Trametinib Restored Osimertinib Sensitivity in NSCLC Cells through ERK/miR-21/PDCD4 Signaling

Since we found that the suppression of Trametinib treatment on tumorgenicity of Osimertinib-resistant P_AZDR1 and H1975+CAF_R1 cells was associated with the decreased miR-21, we further searched for the miR-21′s target to verify the causal relationship. The protein-protein interaction (PPI) network of ERK signaling proteins was evaluated by STRING. Through the network of protein interaction, we found a possible regulatory role of PDCD4 in ERK signaling. PDCD4 is a new type of tumor suppressor that has multiple functions of inhibiting cell growth, tumor invasion, metastasis, and inducing apoptosis. It is down-regulated by microRNA-21 (miR-21) in renal cell carcinoma (RCC) cell lines and tissues. Previous bioinformatics analysis has confirmed that PDCD4 contains the binding site of miR-21 and acts as a tumor suppressor in the regulation of various processes associated with the development of cancer, including cell proliferation, invasion, metastasis, and neoplastic transformation. Based on the above analysis, we hypothesized that the tumor suppressor PDCD4 is one of the effective inhibitory targets of miR-21-5p. (Figure 4A). In a cohort of 511 lung cancer patients, a negative correlation between PDCD4 and miR-21-5p was observed (Figure 4B). In addition, miR-21-5p was found to be able to bind to the 3′UTR of PDCD4 (Insert, Figure 4C). In support, P_AZDR1 and H1975+CAF_R1 cells with silenced miR-21-5p (Inhibitor, I) showed a markedly reduced expression of MEK1/2, STAT3, ERK1/2, and PDCD4 while miR-21-5p overexpressed (Mimic, M) counterparts showed the opposite results (Figure 4C). We also examined the CAF markers in CAF_R1 cells under the influence of miR-21-5p expression. FAP, VIM, α-SAM, and PDGFR in CAF_R1 cells with silenced miR-21-5p were significantly lower (Inhibitor, Figure 4D). The opposite results were noted in the CAF_R1 cells with overexpressed miR-21-5p (Mimic, Figure 4D). The cytokine profiles were also in line with the changes of CAF markers. The CAF_R1 cells having silenced miR-21-5p showed a significantly lower secretion of IL-6, IL-8, and HGF, as compared to their overexpressed miR-21-5p counterparts (Figure 4E). Phenotypically, cells with silenced miR-21-5p showed a significantly lower ability to generate tumor spheres (Inhibitor, Figure 4F) and subsequently increased miR-21-5p by mimic molecules restored this ability (Mimic, Figure 4F). We also show that inhibiting mir-21-5p does the job in commercial cell lines (PC9 and A549, Appendix A). Trametinib is a reversible and highly selective ectopic inhibitor that inhibits the activation and kinase activity of extracellular message-regulated kinase 1 (MEK1) and MEK2 activated by mitogen. MEK protein is an important component of the extracellular message-related kinase (ERK) pathway. In melanoma and other cancers, this pathway is often activated by a mutant form of BRAF, which activates MEK and stimulates tumor cell growth. Trametinib can inhibit the activation of MEK by BRAF and the activity of MEK kinase. Akt and p70S6K are activated by the tumor promoter 12-O-tetradecanoylphorbol-13-acetate (TPA) and are required for the degradation of PDCD4. In addition, the MEK pathway is essential for tumor promoter–induced down-regulation. It is currently known that PDCD4 protein expression is regulated by PI3K/Akt/S6K and MEK/ERK signals. These signals program PDCD4 for ubiquitination and proteasome degradation. The above results imply the inhibition of PDCD4 or the use of MEK inhibitor Trametinib as Treatment may have the same potential effect on MEK/PDCD4 axis. 

### 3.5. In Vivo Evaluation of Therapeutic Effects of Trametinib in Combination with Osimertinib

Next, we evaluated the therapeutic effects of Trametinib by injecting mice with P_AZDR1 cells to create a patient-derived xenograft model. P_AZDR1 bearing mice were divided into 4 groups consisting of vehicle control, mice treated with Trametinib alone (10 mg/kg, P.O, 5 times/week), Osimertinib alone (5 mg/kg, P.O, 5 times/week), and a combination of both drugs (combining both regimens). The flow chart for in vivo experimental design and treatment schedule is shown in Figure 5A. As expected, the Osimertinib group showed a similar tumor growth curve as the vehicle control. Trametinib treatment appeared to significantly be delayed the tumor growth and the combination treatment yielded the most inhibitory effects among all groups (Figure 5B). The longitudinal monitoring of body weight showed no apparent weight loss in the combination group while a slight weight loss (not statistically different) was observed in the vehicle and Osimertinib groups (Figure 5C). In addition, mice with the combination treatment showed the best survival ratio followed by Trametinib alone followed by Osimertinib, and vehicle control (Figure 5D). The qPCR results revealed that the miR-21-5p level was significantly lower in the combination treatment group followed by Trametinib as compared to the samples from the vehicle and Osimertinib groups (Figure 5E). Finally, Immunohistochemical and quantitative analysis indicated that the combination treatment prominently reduced the expression of oncogenic markers including MEK, ERK, STAT3, and PDCD4, as well as increased E-cadherin. Notably, the expression of FAP was also prominently reduced by the combination treatment (Figure 5F).

## 4. Discussion

As one of the main challenges faced by clinicians treating non-small cell lung cancer (NSCLC), drug resistance continues to hinder efforts to slow the progression of the disease. Osimitinib is currently the standard first-line treatment for NSCLC patients with EGFR mutations. In addition, in about half of the first-line Osimertinib failure cases, the molecular mechanism leading to resistance remains unknown. Given Osimertinib resistance, clinicians have explored other strategies, including reusing the first-generation EGFR TKI. In a recent study, Osimertinib plus intermittent Selumetinib (MEK1/MEK2 inhibitor) demonstrated preliminary antitumor activity in patients with EGFR-mutated NSCLC who progressed to prior EGFR TKI. The use of a combination of immune checkpoint inhibitors and tyrosine kinase inhibitors resulted in excessive toxicity without additional efficacy, while immunotherapy and chemotherapy, especially in combination with anti-angiogenic drugs, are more effective than those previously targeted It looks promising for the treated patients. MET and MEK inhibitors may overcome EGFR-independent resistance. Another acquired resistance mechanism of EGFR-TKI is the up-regulation of the RAS/RAF/MEK/ERK signaling pathway, which is the key to cell survival and proliferation; this may occur downstream of various other signaling pathways. In this report, we reveal the possible regulatory mechanism and inhibitory effect of the MEK inhibitor Trametinib applied to MEK/ERK/miR-21 axis and PDCD4 in Osimertinib resistance.

We examined the potential underlying factors contributing to acquired resistance against Osimertinib. Comparatively, we found that clinical samples from Osimertinib-resistant patients contained a significantly higher level of ERK2 expression than the Osimertinib-sensitive counterparts (Figure 1C). This echoes the analysis from the database showing that a higher ERK2 expression was associated with worse survival in lung cancer patients (Figure 1D). We then disclosed the relationship between CAFs and Osimertinib resistance by co-culturing Osimertinib-sensitive NSCLC cells (P_AZDS1 and H1975) with CAF_R cells to increase cell viability under Osimertinib treatment (Figure 2D). This phenomenon was accompanied by the increase of specific marker expression and cytokine release in CAFs (Figure 2A,C), as well as the increase of tumor sphere formation and up-regulation of MEK1/2, ERK1/2, STAT3 plus β-catenin in NSCLC cells (Figure 2E,F). These findings highlighted the critical roles of CAFs and MEK/ERK signaling in Osimertinib resistance. Regarding the roles of CAFs in drug resistance, it has been shown that CAFs secreted IL-6 to enhance epithelial-mesenchymal transition (EMT) and Cisplatin resistance in NSCLC [28]. Though generally considered critical for metastasis, the role of EMT in drug resistance had been described [29]. In addition to promoting EMT, some studies also suggested that the drug-scavenging ability of CAFs may contribute to Gemcitabine failure in pancreatic cancer, while other studies demonstrated that CAFs can inhibit reactive oxygen species (ROS) production to antagonize chemotherapy-induced cell death in prostate cancer [30,31]. Concerning therapeutic strategies aiming at CAFs, it has been reported that the application of inhibitors for STAT3 and MEK1/2 could suppress tumorigenesis in the organotypic model of esophageal cancer [32]. Thus, it is rational to try MEK inhibitors to overcome Osimertinib resistance from the abovementioned results.

We then used MEK inhibitor Trametinib to examine its inhibitory effect on Osimertinib-resistant NSCLC cells (P_AZDR1 and H1975+CAF_R1) and found it restored Osimertinib sensitivity remarkably (Figure 3A). The associated changes in CAFs and NSCLC cells (Figure 3B,D–F) were all varied opposite to th1ose mentioned above accordingly. Meanwhile, the expressions of miR-21 in CAFs and NSCLC cells were found to be parallel to Osimertinib resistance and MEK/ERK expressions (Figure 2B and Figure 3C). Further investigation using mimic and inhibitor confirmed the signaling axis of MEK/ERK/miR-21/PDCD4 (Figure 4). Previously, miR-21 has been well established as an oncomiR in many cancers including lung cancer. It functions as an oncogene through silencing many tumor suppressor genes such as PDCD4, PTEN, SMAD7, HIF-1α, and others, all of which play key roles in different aspects of lung cancer development including cell cycle, apoptosis, drug resistance, and distant metastasis [26,33]. However, direct targeting microRNA as a therapeutic tool has been found to be a challenging task, and indirect suppression of miR-21 seems more feasible. In this study, we found that the treatment of Trametinib was associated with a reduced level of miR-21 in both Osimertinib-resistant NSCLC cells (P_AZDR1 and H1975+CAF_R1) in the concert of increased expression of PDCD4. This provided a partial explanation to the re-sensitization of Osimertinib since PDCD4 is a major tumor suppressor to repress several oncogenic cascades such as c-Myc, Bcl-xL, and mTOR/Akt [34]. It is therefore conceivable that the overtly activated EGFR signaling (T790M+C797S) could be overcome by Trametinib treatment owing to simultaneous inhibition of multiple oncogenic pathways.

MEK is downstream of the RAS-RAF-MEK-ERK pathway (ERK signaling), which had been recognized as the most frequently hyperactivated pathway in human cancers, including NSCLC [35,36]. Clinically, MEK inhibitors have already been applied in several cancers including melanoma, thyroid cancer, and NSCLC [37]. Up to now, there have been four MEK inhibitors approved by the United States Food and Drug Administration (FDA), consisting of Trametinib, Binimetinib, Selumetinib, and Cobimetinib. Among them, only Trametinib was recently approved for the treatment of NSCLC patients in combination with Dabrafenib for those with BRAF V600E mutation [38]. As for MEK inhibitor monotherapy in treating NSCLC patients, most clinical trials were not successful because of poor clinical outcomes and more toxicities as compared with chemotherapy alone [39,40,41]. In the current study, we found combination treatment of Trametinib and Osimertinib could effectively suppress the growth of tumors deriving from P_AZDR1 cells in the xenograft model. Meanwhile, the body weights of the experimental animals were not remarkably decreased after combining two inhibitors, indicating this combination was well tolerated. Our findings provide important and promising preclinical information, but further investigation is warranted for the design and implementation of clinical trials.

A recent report from Gong et al. demonstrated that activated EGFR signaling results in a rapid inhibition of TNF mRNA through miR-21 induction, and inhibition of EGFR results in amplified TNF mRNA and reduced miR-21 expression levels in lung cancer They also showed TNF activated NF-kB, which in turn induces the transcription of TNF mRNA in a feedforward control [42]. In fact, it has been reported earlier that aberrantly increased expression of miR-21 was observed in lung carcinogenesis in smokers and non-smokers, and this is further enhanced by activation of the EGFR signaling pathway [43]. In our current study, we found that CAF_R1 cells had a higher expression of miR-21 (Figure 2B), and it was effectively suppressed by the MEK inhibitor Trametinib (Figure 3C). These were compatible with previous reports because the CAF_R1 cells were derived from P_AZDR1 cells, of which the EGFR signaling was theoretically activated in large amounts due to their characteristics of drug resistance. Regarding the cross-talk between NF-kB and STAT3 signaling, it has been shown in a variety of cancers. NF-kB and STAT3 can bind the same promoters/enhancers and share many downstream genes [44]. The NF-kB/IL-6/STAT3 axis has also been well established [45,46]. STAT3 can also activate NF-kB, but the reported mechanism is still relatively less. Here we identified that the combination treatment prominently reduced the expression of STAT3, which further resulted in the reduced secretion of IL-6. These events might switch off the activation of NF-kB and TNF mRNA feedforward loop and sensitize the NSCLC cells towards the treatment.

Combination treatments have been suggested to provide solutions for NSCLC patients encountering resistance after the administration of target therapies [47,48,49]. The combinations were not limited to target therapy or small molecule therapy but also included chemotherapy and immunotherapy. When focusing on the combination of Osimertinib and MEK inhibitor, there have been some reports addressing this issue. It was reported that this Osimertinib plus Trametinib enhanced induction of apoptosis in EGFR-mutant NSCLC cells but not in EGFR wild-type NSCLC cells, and was very effective in killing cell clones with primary intrinsic resistance to Osimertinib [50]. The same group also reported that the combination of a MEK or ERK inhibitor with a first-generation or second-generation EGFR-TKI also very effectively inhibited the growth of Osimertinib-resistant cells, although these cell lines were cross-resistant to first-generation or second-generation EGFR-TKIs [51]. Our study essentially verified the effectiveness of such a combination, but we also provided evidence of CAF involvement in this process and disclosed the underlying mechanisms focusing on miR-21/PDCD4.

## 5. Conclusions

Cumulatively, as shown in Figure 6, our study provided data that the inhibition of MEK/ERK signaling could overcome Osimertinib resistance both in vitro and in vivo. Mechanistically, MEK inhibitor Trametinib suppressed tumorigenic properties in NSCLC cells and also in turn reduced the generation of CAFs. The Trametinib-mediated anti-cancer function was associated with a significantly suppressed level of miR-21, of which primary targets included PDCD4, as shown in this study. 

## Figures and Tables

**Figure 1 cancers-13-06005-f001:**
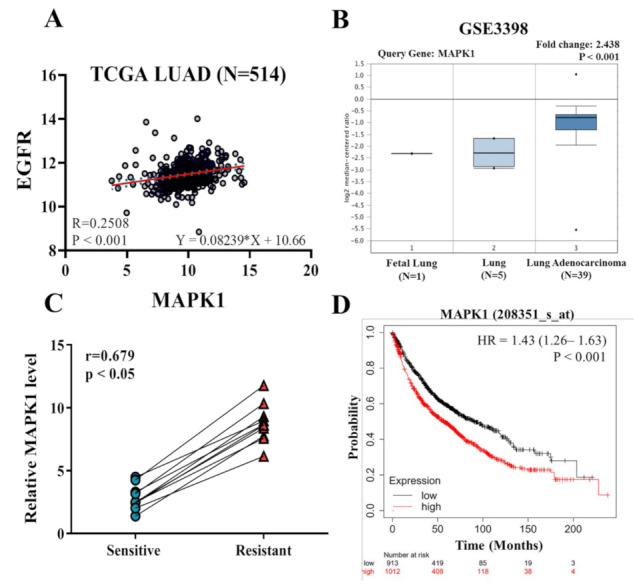
Elevated co-expression of EGFR and MAPK1 (also known as ERK2) is associated with poor prognosis in NSCLC patients and Osimertinib resistance. (**A**) TCGA lung adenocarcinoma database analysis (N = 514) revealed a positive correlation between EGFR and ERK (r = 0.2508, *p* < 0.001). (**B**) Comparative expression analysis shows a significantly higher MAPK1 mRNA in lung adenocarcinoma samples as compared to normal lung and fetal lung tissues (*p* < 0.001) [16]. (**C**) Clinical samples of Osimertinib-resistant NSCLC samples contained a significantly higher MAPK1 level as compared to their sensitive counterparts (10 pairs). Pearson correlation r = 0.679, *p* < 0.05. (**D**) Kaplan-Meier analysis of the NSCLC cohort (N = 1925) revealed a significant correlation between an elevated MAPK1 mRNA with worse survival (log-rank *p* < 0.001).

**Figure 2 cancers-13-06005-f002:**
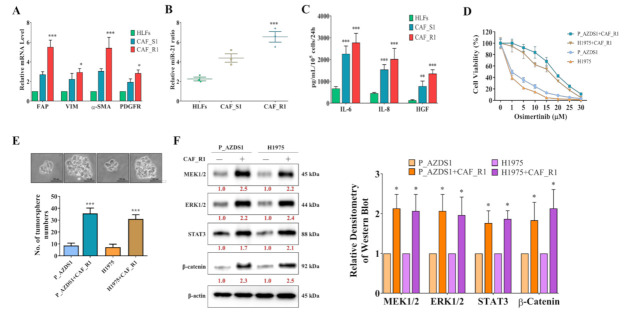
Osimertinib-resistant NSCLC cells transformed normal fibroblasts to cancer-associated fibroblasts and promoted drug resistance. (**A**) Real-time PCR analysis of CAFs transformed by co-culture with NSCLC cells. CAF markers were significantly higher in the CAFs when co-cultured with Osimertinib-resistant P_AZDR1 cells (CAF_R1 cells) as compared to Osimertinib-sensitive P_AZDS1 cells (CAF_S1 cells). (**B**) qPCR results showed that CAF_R1 cells contained a significantly higher level of oncomir, miR-21 than HLF and CAF_S1. (**C**) Comparative CAF cytokine profiles showed that CAF_R1 cells secreted a significantly higher IL-6, IL-8, and HGF than CAF_S1 and HLF. a, * *p* < 0.05; b, ** *p* < 0.01; c, *** *p* < 0.001 (**D**) Cell viability assay showed that CAF_R1 cells co-culture led to an increased Osimertinib resistance in both P_AZDS1 and H1975 NSCLC cells. (**E**) Co-culturing with CAF_R1 cells resulted in significantly increased tumor sphere formation ability in both P_AZDS1 and H1975 cells. (**F**) CAF_R1 cells co-cultured P_AZDS1 and H1975 cells showed a prominently increased expression of MEK, ERK, STAT3, and β-catenin as compared to their naïve counterparts and bar graph of multiple experiments. ** *p* < 0.01, *** *p* < 0.001. Scale bar: 100 μm.

**Figure 3 cancers-13-06005-f003:**
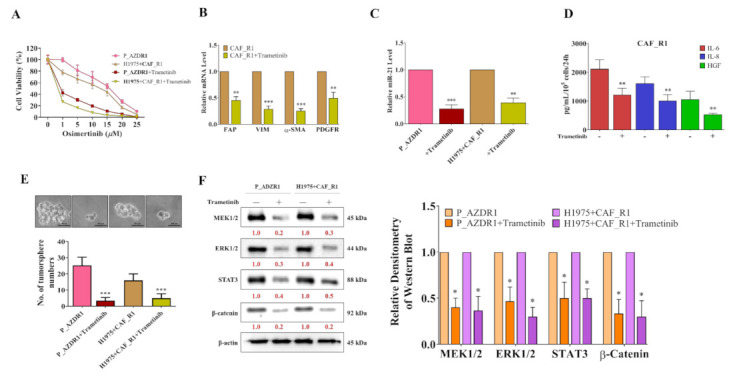
(**A**) Cell viability assay showed that sequential treatment of Trametinib (0.1 uM, 24 h) and Osimertinib (2 uM, 48 h) re-sensitized Osimertinib-resistant NSCLC cells (P_AZDR1 and H1975+CAF_R1). (**B**) Trametinib treatment resulted in significantly reduced CAF markers including FAP, VIM, α-SMA, and PDGFR in CAF_R1 cells. (**C**) qPCR analysis showed that Trametinib-treated P_AZDR1 and H1975+CAF_R1 cells contained a significantly lower level of miR-21 level. (**D**) ELISA assay analysis showed that CAF-associated cytokines, IL-6, IL-8, and HGF released were significantly reduced by the treatment of Trametinib in CAF_R1 cells. (**E**) Tumor sphere formation assay indicated that Trametinib treatment suppressed the self-renewal ability of both P_AZDR1 and H1975+CAF_R1 cells significantly and was supported by the western blot (**F**) where Trametinib-treated cells expressed a markedly reduced level of MEK1/2, ERK1/2, STAT3 and β-catenin and bar graph of multiple experiments. * *p* < 0.05, ** *p* < 0.01, *** *p* < 0.001. Scale bar: 100 μm.

**Figure 4 cancers-13-06005-f004:**
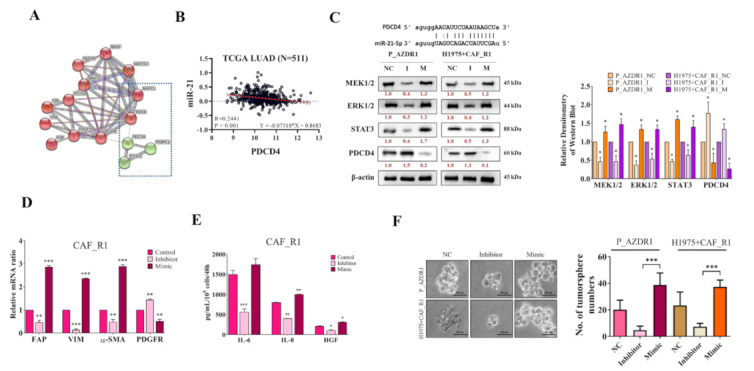
(**A**) Schematic representation of the interaction of the ERK/miR-21 signaling proteins by STRING analysis and PDCD4 was identified. (**B**) TCGA database analysis of 512 lung cancer patients showed a negative correlation between the expression of miR-21-5p and PDCD4. (**C**) PDCD4 was predicted as a target of miR-21-5p. The insert shows the miR-21-5p binds to the 3′-UTR of PDCD4. Western blots showed a decreased expression of MEK1/2, STAT3, ERK1/2, and PDCD4 in miR-21-5p inhibitor (I) treated lung cancer cells and the overexpression of miR-21-5p (M) showed the reverse and bar graph of multiple experiments. (**D**) Real-time PCR analysis showed that CAF_R1 cells with silenced miR-21-5p expressed a significantly lower level of FAP, A-SAM, VIM, and PDGFR. (**E**) CAF-associated cytokines, IL-6, IL-8, and HGF were also significantly reduced in the CAF_R1 cells with silenced miR-21-5p. (**F**) Tumor sphere formation assay. In both miR-21-5p inhibitor-treated P_AZDR1 and H1975-CAF_R1 cells, a significantly lower number of spheres was formed while mimic treatment reversed this phenomenon. * *p* < 0.05; ** *p* < 0.01; *** *p* < 0.001. Scale bar: 100 μm.

**Figure 5 cancers-13-06005-f005:**
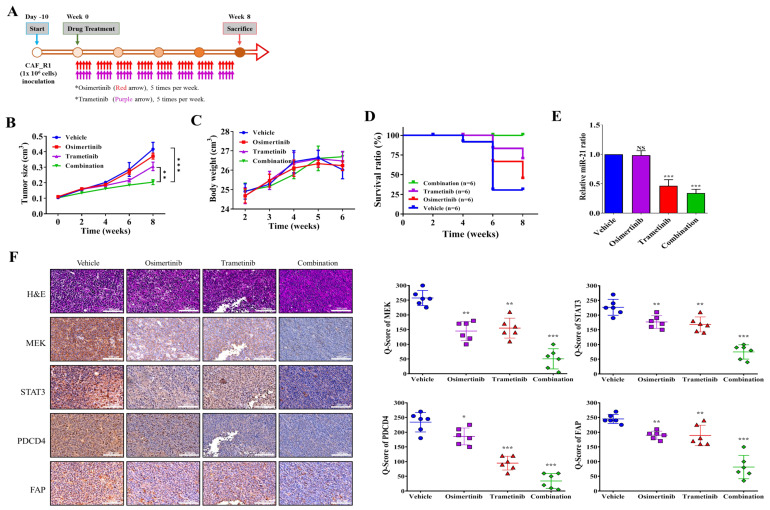
Combined treatment of Trametinib and Osimertinib effectively inhibited the growth of P_AZDR1 (Osimertinib resistant) cells in vivo. (**A**) The flow chart for in vivo experimental design and treatment schedule. (**B**) Tumor curve over time shows that the combination of Trametinib and Osimertinib suppressed the tumor growth the most followed by the Trametinib alone group as compared to the vehicle and Osimertinib groups (no significant difference between these two groups). (**C**) Bodyweight over time curve demonstrated no apparent systematic toxicity in the mice receiving combined treatment. (**D**) Kaplan-Meier survival curve shows that the mice that received the combined treatment exhibited the highest survival ratio as compared to the rest. (**E**) Real-time PCR results of the tumor samples revealed that the miR-21-5p level was most significantly suppressed in the combined treatment followed by the Trametinib group while no significant difference was observed between vehicle and Osimertinib groups. (**F**) Immunostaining analysis of PDX tumor sections showed that the combined treatment most prominently suppressed the expression of MEK, ERK, STAT3 and PDCD4 but an increased E-cadherin was observed, compared to other sections. * *p* < 0.05; ** *p* < 0.01; *** *p* < 0.001. NS, not significantly different. PDX: patient-derived xenograft.

**Figure 6 cancers-13-06005-f006:**
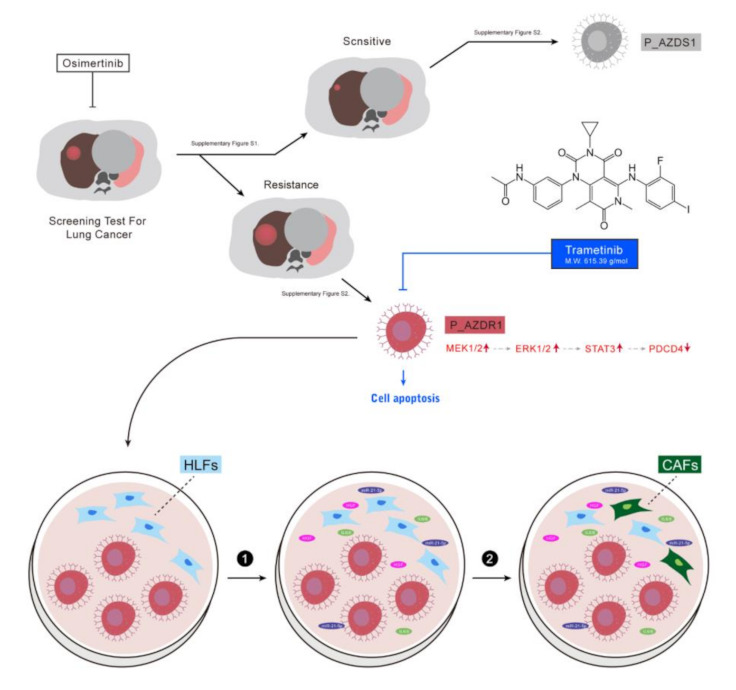
EGFR activating mutation advanced with T790M/C797S mutation led to an amplification of MEK/ERK signaling resulting in survival and resistance against Osimertinib. Osimertinib-resistant NSCLC cells promoted the transformation of human lung fibroblasts (HLF) to cancer-associated fibroblasts (CAFs) via IL-6/STAT3 signaling; in turn, CAFs promoted tumorigenesis by secreting IL-6, IL-8, and HGF as well as increasing oncomiR miR-21 in NSCLC cells. MEK inhibitor, Trametinib treatment suppresses MEK/ERK signaling to render Osimertinib-resistant NSCLC sensitive to treatment.

## Data Availability

The datasets used and analyzed in the current study are publicly accessible as indicated in the manuscript.

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
