# Peer review of "The MEK/ERK/miR-21 Signaling Is Critical in Osimertinib Resistance in EGFR-Mutant Non-Small Cell Lung Cancer Cells"

_cancers, 2021, doi:10.3390/cancers13236005_

Round 1
Reviewer 1 Report
Dr Huang and colleagues in the manuscript entitled “The MEK/ERK/miR-21 signaling is critical in Osimertinib resistance in EGFR-mutant non-small cell lung cancer cells” started from a TGCA analysis exhibiting a correlation between ERK2 levels and both osimertinib resistance and poor prognosis in NSCLC patients. They evaluated the contribution of CAFs to such a resistance and found that osimertinib resistant NSCLC cells do instruct CAFS toward a protumorigenic phenotype in terms of cytokine secretion and expression of EMT markers. They showed that the instructed CAFs bore increased ERK and STAT3 levels when mixed with epithelial lung cancer cells and were endowed with higher levels of mir21. Finally, they identified PDCD4 as target of mir-21 and showed that ERK inhibition by trametinib may counteract the mir-21 protumorigenic activity toward PDCD4 in a clinically relevant model.
This is an interesting work, which makes use of an effective mix of experimental approaches, ranging from in vitro assay to in vivo evaluation of one PDX.
There is already a number of reports suggesting a role for MEK/ERK in mediating resistance to osimertinib (https://doi.org/10.1002/cncr.32996) or showing the effectiveness of combining osimertinib with trametininb and occasionally dabrafenib (when the BRAF V660 mutation arises as in osimertinib treated patients) (DOI: 10.1016/j.lungcan.2020.05.036), in EGFRmut NSCLC. From such a point of view, this work is a late comer. However, the suggested MoA is quite novel and may add potential for studying some of the identified targets (mir-21-5p, PDCD4), for patient stratification purposes. The figures are rich but self-explanatory, the methods are appropriately detailed and the statistical analysis is acceptable. Generally, the data support the authors’ conclusions. The text is generally clear but would ultimately benefit of a revision.
Issues. There is number of issues, which need to be addressed in order for this work to fully unleash its translational potential.
General. This is a data-rich work, however, the logical connection between the experiments performed needs a deeper and more detailed explanation. Here below two examples, not exhaustive of all the other passages.
Lines 324-328. “Since we found that the suppression of Trametinib treatment on tumorgenicity of Osimertinib-resistant P_AZDR1 and H1975+CAF_R1 cells was associated with the decreased miR-21, we further searched for miR-21’s target to verify the causal relationship.” It is unclear what the authors refer to. Is it already published work? There is no reference to it and no reference to any figure of the manuscript (is it figure 3A-F?) Please detail since this is very relevant to explain the experiments performed and to understand the logical flow of the experiments.
Another example. It is likely that the authors decided to employ trametinib treatment after observing ERK1/2 reduced in mir-21-5p inhibitor-treated cells. The hypothesis could have been that, at least partially, the trametinib treatment may mimic the observed effects of mir-21-5p inhibition. The authors need to state whether this was the working hypothesis and need to explain and argument this both in the results and in the discussion section. Otherwise, the whole experimental process remains unclear.
In line with these remarks, we believe the authors should reformat the abstract and partly the introduction to better evidence the logic they have followed. Since MEK/ERK signaling was already described as one of the mechanisms involved in Osimertinib resistance, what is the specific background to the hierarchy of their experiments and demonstrations? Which is the logic argumentation for the single steps of the demonstration?
Other issues.
Figure 3F. The figure reveals effective target engagement by trametinib. Please provide a quantification of multiple western blotting experiments. In the 3F, the migration of the beta catenin stained bands is not coherent with the rest of the gel (even after looking at the supplementary figures). And, also, the migration of STAT3 is altered. Can the authors provide a better representative western blotting image? Can the authors clarify and describe in the legend whether the results in 3F (and 2F as well) derive from multiple, collated gels?
Figure 3A: what happens to the number of CAFs when the coculture is treated with trametininb? The decrease in mCAF- associated cytokines and markers may be explained by a selective targeting of CAFs by the trametinib. The authors need to investigate mechanistically on this to ensure a clearer picture and further support their data.
Line 332-334 and figure 4C. The authors report that inhibiting mir-21-5p leads to decreased PDCD4 levels. This is not what observed in figure 4C. Given the tumor suppressor role of PDCD4, such a claim would have been counter logical. Please correct.
Figure 4C. The effect of the mir-21.5p mimic is not very clear. In order to better support the claim that mir-21-5p mimic reduces the levels of PDCD4, the authors may show additional western blotting and report the findings of multiple experiments as a bar graph.
Figure 4F (and figure 3F). The lungsphere formation assay may not be appropriate. Firstly, it looks like that, rather than spheres, floating aggregates were generated. Secondly, it is important to understand whether the inhibited aggregate formation comes from a reduced viability after inhibitor treatment. The authors should use, if possible, different cells (even a commercial one) to show that inhibiting mir-21-5p does the job. Additionally, a proper Sphere forming assay entails plating the cells at different dilutions and counting the number of formed spheres, possibly in a 96 well format. Also measuring the diameter of the formed spheres would complement such experiment.
Figure 5F. Please provide quantitative analysis of the IHC data. This would both clarify and further support the authors’ conclusions.
Author Response
We thank the reviewer for carefully reading our manuscript and providing valuable comments. We accordingly response the questions raised by the Reviewer as follows:
Response to Reviewers:
Reviewer #1 (Comments to the Author):
Q1: There is already a number of reports suggesting a role for MEK/ERK in mediating resistance to osimertinib (http://sci-hub.tw/10.1002/cncr.32996) or showing the effectiveness of combining osimertinib with trametininb and occasionally dabrafenib (when the BRAF V660 mutation arises as in osimertinib treated patients) (DOI: 10.1016/j.lungcan.2020.05.036), in EGFRmut NSCLC. From such a point of view, this work is a late comer. However, the suggested MoA is quite novel and may add potential for studying some of the identified targets (mir-21-5p, PDCD4), for patient stratification purposes. The figures are rich but self-explanatory, the methods are appropriately detailed and the statistical analysis is acceptable. Generally, the data support the authors’ conclusions. The text is generally clear but would ultimately benefit of a revision. Issues. There is number of issues, which need to be addressed in order for this work to fully unleash its translational potential. General. This is a data-rich work, however, the logical connection between the experiments performed needs a deeper and more detailed explanation. Here below two examples, not exhaustive of all the other passages. Lines 324-328. “Since we found that the suppression of Trametinib treatment on tumorgenicity of Osimertinib-resistant P_AZDR1 and H1975+CAF_R1 cells was associated with the decreased miR-21, we further searched for miR-21’s target to verify the causal relationship.” It is unclear what the authors refer to. Is it already published work? There is no reference to it and no reference to any figure of the manuscript (is it figure 3A-F?) Please detail since this is very relevant to explain the experiments performed and to understand the logical flow of the experiments. Another example. It is likely that the authors decided to employ trametinib treatment after observing ERK1/2 reduced in mir-21-5p inhibitor-treated cells. The hypothesis could have been that, at least partially, the trametinib treatment may mimic the observed effects of mir-21-5p inhibition. The authors need to state whether this was the working hypothesis and need to explain and argument this both in the results and in the discussion section. Otherwise, the whole experimental process remains unclear.
A1. We sincerely thank the reviewer for this insightful suggestion. In the revised manuscript, we added the reason for searching for miR-21 targets in the manuscript to help readers understand the logic of experimental design. Please kindly see our revised result section.
“1. Introduction
…..Nevertheless, similar to previous generations of TKIs, NSCLC patients eventually develop resistance against Osimertinib, via EGFR-dependent and EGFR-independent pathways. The EGFR-dependent pathway includes the acquisition of C797S mutation [12,13], as well as other less frequent mutations such as G796 and L792 [14,15], on the kinase-binding site of EGFR. The EGFR-independent pathway includes activation of other receptor tyrosine kinases (RTKs) which share the common downstream pathways of EGFR, like HER2 and MET [16,17], or working via other mutated genes, for example, RAS and BRAF V600E [18,19]. Based on these clinical observations, it appears that TKI treatment inevitably drives the tumor cells to adapt to the treatment and acquire resistance. Thus, the development of alternative strategies which take into account multiple targets should be considered. Tumor microenvironment (TME) has been well recognized as an essential component for tumorigenesis. One of the major players within the tumor microenvironment is the cancer-associated fibroblasts (CAFs). CAFs are one of the most abundant components in the TME and have the ability to promote tumor progression. CAFs promote angiogenesis and migration by synthesizing favourable extracellular matrix (ECM) [20,21], and help immune evasion by recruiting immunosuppressive cells via the production of growth factors and inflammatory cytokines [22,23]. Meanwhile, HGF (hepatocyte growth factor) produced by CAFs, in turn, activates the MAPK and PI3K/AKT pathways in the tumor cells, leading to resistance to the treatments [24,25]. Collectively, tumor cells and CAFs have intimate crosstalk to promote cancer progression and targeting tumor cells alone may be insufficient. Therefore, the elimination of CAFs should also be considered when constructing therapeutic interventions. Meanwhile, with the growing evidence of the role of small non-coding RNA, particularly micro-RNA (miRNA), in altering and regu-lating many key biological processes of various cancers, the important role of miR-21 in lung cancer has also been addressed [26].”
“3.4. Trametinib restored Osimertinib sensitivity in NSCLC cells through ERK/miR-21/PDCD4 signaling
Since we found that the suppression of Trametinib treatment on tumorgenicity of Osimertinib-resistant P_AZDR1 and H1975+CAF_R1 cells was associated with the decreased miR-21, we further searched for miR-21’s target to verify the causal relationship. The protein-protein interaction (PPI) network of ERK signaling proteins was evaluated by STRING.Through the network of protein interaction, we found a possible regulatory role of PDCD4 in ERK signaling. PDCD4 is a new type of tumor suppressor that has multiple functions of inhibiting cell growth, tumor invasion, metastasis, and inducing apoptosis. It is down-regulated by microRNA-21 (miR-21) in renal cell carcinoma (RCC) cell lines and tissues. Previous bioinformatics analysis has confirmed that PDCD4 contains the binding site of miR-21 and acts as a tumor suppressor in the regulation of various processes associated with the development of cancer, including cell proliferation, invasion, metastasis, and neoplastic transformation. Based on the above analysis, we hypothesized that the tumor suppressor PDCD4 is one of the effective inhibitory targets of miR-21-5p. (Figure 4A). In a cohort of 511 lung cancer patients, a negative correlation between PDCD4 and miR-21-5p was observed (Figure 4B). In addition, miR-21-5p was found to be able to bind to the 3’UTR of PDCD4 (Insert, Figure 4C). In support, P_AZDR1 and H1975+CAF_R1 cells with silenced miR-21-5p (Inhibitor, I) showed a markedly reduced expression of MEK1/2, STAT3, ERK1/2 and PDCD4 while miR-21-5p overexpressed (Mimic, M) counterparts showed the opposite results (Figure 4C). We also examined the CAF markers in CAF_R1 cells under the influence of miR-21-5p expression. FAP, VIM, α-SAM and PDGFR in CAF_R1 cells with silenced miR-21-5p were significantly lower (Inhibitor, Figure 4D). The opposite results were noted in the CAF_R1 cells with overexpressed miR-21-5p (Mimic, Figure 4D). The cytokine profiles were also in line with the changes of CAF markers. The CAF_R1 cells having silenced miR-21-5p showed a significantly lower secretion of IL-6, IL-8 and HGF, as compared to their overexpressed miR-21-5p counterparts (Figure 4E). Phenotypically, cells with silenced miR-21-5p showed a significantly lower ability to generate tumor spheres (Inhibitor, Figure 4F) and subsequently increased miR-21-5p by mimic molecules restored this ability (Mimic, Figure 4F). Trametinib is a reversible and highly selective ectopic inhibitor that inhibits the activation and kinase activity of extracellular message-regulated kinase 1 (MEK1) and MEK2 activated by mitogen. MEK protein is an important component of the extracellular message-related kinase (ERK) pathway. In melanoma and other cancers, this pathway is often activated by a mutant form of BRAF, which activates MEK and stimulates tumor cell growth. Trametinib can inhibit the activation of MEK by BRAF and the activity of MEK kinase.Akt and p70S6K are activated by the tumor promoter 12-O-tetradecanoylphorbol-13-acetate (TPA) and are required for degradation of PDCD4. In addition, MEK pathway is essential for tumor promoter–induced down-regulation.It is currently known that PDCD4 protein expression is regulated by PI3K/Akt/S6K and MEK/ERK signals. These signals program PDCD4 for ubiquitination and proteasome degradation. The above results imply the inhibition of PDCD4 or the use of MEK inhibitor Trametinib as Treatment may have the same potential effect on MEK/ PDCD4 axis.”
Q2: In line with these remarks, we believe the authors should reformat the abstract and partly the introduction to better evidence the logic they have followed. Since MEK/ERK signaling was already described as one of the mechanisms involved in Osimertinib resistance, what is the specific background to the hierarchy of their experiments and demonstrations? Which is the logic argumentation for the single steps of the demonstration?
A2. We sincerely thank the reviewer for this insightful suggestion. In the revised manuscript, we have reformat the abstract and partly the introduction to better evidence the logic they have followed. Please kindly see our revised Abstract section.
“Abstract:
Background: The third-generation epidermal growth factor receptor (EGFR) inhibitor, Osimertinib, is used to treat non-small cell lung cancer (NSCLC) patients with tyrosine kinase in-hibitor (TKI) resistance caused by acquired EGFR T790M mutation. However, patients eventually develop resistance against Osimertinib with mechanisms not yet fully clarified. Activated alterna-tive survival pathways within the tumor cells and cancer-associated fibroblasts (CAFs) have been proposed to contribute to Osimertinib resistance. MET and MEK inhibitors may overcome EGFR-independent resistance. Another acquired resistance mechanism of EGFR-TKI is the up-regulation of the RAS/RAF/MEK/ERK signaling pathway, which is the key to cell survival and proliferation; this may occur downstream of various other signaling pathways.In this report, we reveal the possible regulatory mechanism and inhibitory effect of the MEK inhibitor trametininb applied to MEK/ERK/miR-21 axis and PDCD4 in Osimertinib resistance.We found a possible regulatory role of PDCD4 in ERK signaling.PDCD4 is a new type of tumor suppressor that has multiple functions of inhibiting cell growth, tumor invasion, metastasis, and inducing apoptosis. Previous bioinformatics analysis has confirmed that PDCD4 contains the binding site of miR-21 and acts as a tumor suppressor in the regulation of various processes associated with the development of cancer, including cell proliferation, invasion, metastasis, and neoplastic transformation. Based on the above analysis, we hypothesized that the tumor suppressor PDCD4 is one of the effective inhibitory targets of miR-21-5p. Methods: The expression between EGFR and ERK2 in lung adenocarcinoma was evaluated from the TCGA database. Osimertinib-sensitive and resistant NSCLC cells obtained from patients were used to co-culture with human lung fibroblasts (HLFs) to generate CAF cells (termed CAF_R1 and CAF_S1), and the functional roles of these CAF cells plus the regulatory mechanisms were further explored. Then MEK inhibitor Trametinib with or without Osimertinib was applied in xenograft model derived from patients to verify the effects on growth inhibition of Osimertinib-resistant NSCLC tumors. Result: ERK2 expression correlated with EGFR expression and higher ERK2 level was associated with worse prognosis of patients and Osimertinib resistance. CAFs derived from Osimertinib-resistant cells secreted more IL-6, IL-8 and hepatocyte growth factor (HGF), expressed stronger CAF markers including α-smooth muscle actin (α-SMA), fibroblast activation protein (FAP) plus platelet-derived growth factor receptor (PDGFR), and enhanced stemness and Osimertinib resistance in NSCLC cells. Meanwhile, increased MEK/ERK/miR-21 expressions were found in both CAFs and NSCLC cells. MEK inhibitor Trametinib significantly abrogated the abovementioned effects by modulating β-catenin, STAT3 and ERK. The xenograft model showed combining Osimertinib and Trametinib resulted in the most prominent growth inhibition of Osimertinib-resistant NSCLC tumors. Conclusion: Our results suggested that MEK/ERK/miR-21 signaling is critical in Osimertinib resistance and CAF transformation of NSCLC cells, and MEK inhibitor Trametinib significantly suppressed Osimertinib-resistant NSCLC tumor growth by abolishing both processes.”
“1. Introduction
…..Tumor microenvironment (TME) has been well recognized as an essential component for tumorigenesis. One of the major players within the tumor microenvironment is the cancer-associated fibroblasts (CAFs). CAFs are one of the most abundant components in the TME and have the ability to promote tumor progression. CAFs promote angiogenesis and migration by synthesizing favourable extracellular matrix (ECM) [20,21], and help immune evasion by recruiting immunosuppressive cells via the production of growth factors and inflammatory cytokines [22,23]. Meanwhile, HGF (hepatocyte growth factor) produced by CAFs, in turn, activates the MAPK and PI3K/AKT pathways in the tumor cells, leading to resistance to the treatments [24,25]. Collectively, tumor cells and CAFs have intimate crosstalk to promote cancer progression and targeting tumor cells alone may be insufficient. Therefore, the elimination of CAFs should also be considered when constructing therapeutic interventions. Meanwhile, with the growing evidence of the role of small non-coding RNA, particularly micro-RNA (miRNA), in altering and regu-lating many key biological processes of various cancers, the important role of miR-21 in lung cancer has also been addressed [26].
In this study, we first used the TCGA lung adenocarcinoma database to recognize the correlation between EGFR and ERK expression and evaluated the association between ERK and Osimertinib resistance in tissue pairs. Then we induced CAFs by co-culturing NSCLC cells deriving from patients of different Osimertinib sensitivity with human lung fibroblasts (HLF) and investigated the characteristics CAFs and NSCLC cells. The role of MEK/ERK/miR-21 pathway was further investigated. We also applied Trametinib (a MEK1/2 inhibitor) to verify the suppressive effects on tumorigenesis of NSCLCs and CAFs. Finally, using a patient-derived xenograft model, we demonstrated the efficacy of Trametinib in combination with Osimertinib to inhibit the growth of Osimertinib-resistant cancer cells in vivo.”
Other issues.
Q3: Figure 3F. The figure reveals effective target engagement by trametinib. Please provide a quantification of multiple western blotting experiments. In the 3F, the migration of the beta catenin stained bands is not coherent with the rest of the gel (even after looking at the supplementary figures). And, also, the migration of STAT3 is altered. Can the authors provide a better representative western blotting image? Can the authors clarify and describe in the legend whether the results in 3F (and 2F as well) derive from multiple, collated gels?
A3: We sincerely thank the reviewer for this insightful suggestion. In the revised manuscript, we have provided a quantification of multiple western blotting, include all parts mentioned by the reviewer. We rerun the PAGE and confirmed the above problem. Please kindly see our revised result section.
“Figure 3. (A) Cell viability assay showed that sequential treatment of Trametinib (0.1uM, 24h) and Osimertinib (2uM, 48h) re-sensitized Osimertinib-resistant NSCLC cells (P_AZDR1 and H1975+CAF_R1). (B) Trametinib treatment result-ed in significantly reduced CAF markers including FAP, VIM, α-SMA and PDGFR in CAF_R1 cells. (C) qPCR analysis showed that Trametinib-treated P_AZDR1 and H1975+CAF_R1 cells contained a significantly lower level of miR-21 level. (D) ELISA assay analysis showed that CAF-associated cytokines, IL-6, IL-8 and HGF released were significantly reduced by the treatment of Trametinib in CAF_R1 cells. (E) Tumor sphere formation assay indicated that Trametinib treatment suppressed the self-renewal ability of both P_AZDR1 and H1975+CAF_R1 cells significantly and was sup-ported by the western blot (F) where Trametinib-treated cells expressed a markedly reduced level of MEK1/2, ERK1/2, STAT3 and β-catenin and bar graph of multiple experiments. Scale bar: 100 μm.”
Q4: Figure 3A: what happens to the number of CAFs when the co culture is treated with trametininb? The decrease in mCAF- associated cytokines and markers may be explained by a selective targeting of CAFs by the trametinib. The authors need to investigate mechanistically on this to ensure a clearer picture and further support their data.
A4: We sincerely thank the reviewer for this insightful suggestion. In the revised manuscript, we have check the number of CAFs when the coculture is treated with trametininb. We provide a clearer picture and further support our data. Please kindly see our revised result section.
“3.3. Trametinib significantly reduced tumorigenic properties of Osimertinib-resistant NSCLC cells
In order to confirm the change in the amount of CAF when trametinb is co-cultured, we evaluated the potential therapeutic effects of MEK inhibitor Trametinib for overcoming Osimertinib resistance using P_AZDR1 and H1975+CAF_R1 cells (co-cultured with CAF_R1). Assays for cell viability may monitor the number of cells over time, the number of cellular divisions, metabolic activity, or DNA synthesis. We first observed that Trametinib treatment (0.1μM, 24h) followed by Osimertinib (2μM, 48h) significantly reduced the cell viability of both P-AZDR1 and H1975+CAF_R1 cells (Figure 3A) with the respective IC50 values of Osimertinib as 0.5µM and 0.25 µM, respectively. Equally important, a low concentration of Trametinib (0.1μM) significantly reduced the mRNA level of FAP, VIM, α-SMA and PDGFR in CAF_R1 cells (Figure 3B), indicative of reduced CAF phenotypic presentations. Consistently, the miR-21 level was significantly lowered in Trametinib-treated P_AZDR1 and H1975+CAF_R1 cells (Figure 3C). CAF-associated cytokines, including IL-6, IL-8 and HGF, were also reduced in the culture medium after Trametinib treatment (0.1µM, 24h, Figure 3D). Notably, Trametinib-treated P_AZDR1 and H1975+CAF_R1 cells were significantly less capable of generating tumor spheres as compared with the non-treatment counterparts (Figure 3E). Western blots of lysates from Trametinib treated P_AZDR1 and H1975+CAF_R1 cells showed a prominently suppressed level of MEK1/2, ERK1/2, STAT3 and β-catenin (Figure 3F).”
“Figure 3. (A) Cell viability assay showed that sequential treatment of Trametinib (0.1uM, 24h) and Osimertinib (2uM, 48h) re-sensitized Osimertinib-resistant NSCLC cells (P_AZDR1 and H1975+CAF_R1). (B) Trametinib treatment result-ed in significantly reduced CAF markers including FAP, VIM, α-SMA and PDGFR in CAF_R1 cells. (C) qPCR analysis showed that Trametinib-treated P_AZDR1 and H1975+CAF_R1 cells contained a significantly lower level of miR-21 level. (D) ELISA assay analysis showed that CAF-associated cytokines, IL-6, IL-8 and HGF released were significantly reduced by the treatment of Trametinib in CAF_R1 cells. (E) Tumor sphere formation assay indicated that Trametinib treatment suppressed the self-renewal ability of both P_AZDR1 and H1975+CAF_R1 cells significantly and was sup-ported by the western blot (F) where Trametinib-treated cells expressed a markedly reduced level of MEK1/2, ERK1/2, STAT3 and β-catenin and bar graph of multiple experiments. Scale bar: 100 μm.”
Q5: Line 332-334 and figure 4C. The authors report that inhibiting mir-21-5p leads to decreased PDCD4 levels. This is not what observed in figure 4C. Given the tumor suppressor role of PDCD4, such a claim would have been counter logical. Please correct.
A5: We sincerely thank the reviewer for this insightful suggestion. In the revised manuscript, we reconfirmed and corrected this logic. We rerun the PAGE and confirmed the above problem. Please kindly see our revised result section.
Q6: Figure 4C. The effect of the mir-21.5p mimic is not very clear. In order to better support the claim that mir-21-5p mimic reduces the levels of PDCD4, the authors may show additional western blotting and report the findings of multiple experiments as a bar graph.
A6. We sincerely thank the reviewer for this insightful suggestion. In the revised manuscript. In order to better support the claim that mir-21-5p mimic reduces the levels of PDCD4, we have showed additional western blotting and report the findings of multiple experiments as a bar graph. We also rerun the PAGE and confirmed the above problem. Please kindly see our revised result section.
“Figure 4. (A) Schematic representation of the interaction of the ERK/miR-21 signaling proteins by STRING analysis and PDCD4 was identified. (B) TCGA database analysis of 512 lung cancer patients showed a negative correlation between the expression of miR-21-5p and PDCD4. (C) PDCD4 was predicted as a target of miR-21-5p. The insert shows the miR-21-5p binds to the 3’UTR of PDCD4. Western blots showed a decreased expression of MEK1/2, STAT3, ERK1/2 and PDCD4 in miR-21-5p inhibitor (I) treated lung cancer cells and the overexpression of miR-21-5p (M) showed the reverse and bar graph of multiple experiments. (D) Real-time PCR analysis showed that CAF_R1 cells with silenced miR-21-5p expressed a significantly lower level of FAP, A-SAM, VIM and PDGFR. (E) CAF-associated cytokines, IL-6, IL-8 and HGF were also significantly reduced in the CAF_R1 cells with silenced miR-21-5p. (F) Tumor sphere formation assay. In both miR-21-5p inhibitor-treated P_AZDR1 and H1975+CAF_R1 cells, a significantly lower number of spheres was formed while mimic treatment reversed this phenomenon. **P<0.01; ***P<0.001. Scale bar: 100 μm.”
Q7: Figure 4F (and figure 3F). The lungsphere formation assay may not be appropriate. Firstly, it looks like that, rather than spheres, floating aggregates were generated. Secondly, it is important to understand whether the inhibited aggregate formation comes from a reduced viability after inhibitor treatment. The authors should use, if possible, different cells (even a commercial one) to show that inhibiting mir-21-5p does the job. Additionally, a proper Sphere forming assay entails plating the cells at different dilutions and counting the number of formed spheres, possibly in a 96 well format. Also measuring the diameter of the formed spheres would complement such experiment.
A7. We sincerely thank the reviewer for this insightful suggestion. In the revised manuscript, we have added the relevant data and presentation of the Sphere forming assay mentioned by the reviewer. Please kindly see our revised result section.
“3.4. Trametinib restored Osimertinib sensitivity in NSCLC cells through ERK/miR-21/PDCD4 signaling
….Phenotypically, cells with silenced miR-21-5p showed a significantly lower ability to generate tumor spheres (Inhibitor, Figure 4F) and subsequently increased miR-21-5p by mimic molecules restored this ability (Mimic, Figure 4F). We also show that inhibiting mir-21-5p does the job in commercial cell lines (PC9 and A549, Supplementary S3). Trametinib is a reversible and highly selective ectopic inhibitor that inhibits the activation and kinase activity of extracellular message-regulated kinase 1 (MEK1) and MEK2 activated by mitogen. MEK protein is an important component of the extracellular message-related kinase (ERK) pathway. In melanoma and other cancers, this pathway is often activated by a mutant form of BRAF, which activates MEK and stimulates tumor cell growth. Trametinib can inhibit the activation of MEK by BRAF and the activity of MEK kinase.Akt and p70S6K are activated by the tumor promoter 12-O-tetradecanoylphorbol-13-acetate (TPA) and are required for degradation of PDCD4. In addition, MEK pathway is essential for tumor promoter–induced down-regulation.It is currently known that PDCD4 protein expression is regulated by PI3K/Akt/S6K and MEK/ERK signals. These signals program PDCD4 for ubiquitination and proteasome degradation. The above results imply the inhibition of PDCD4 or the use of MEK inhibitor Trametinib as Treatment may have the same potential effect on MEK/ PDCD4 axis.”
“Figure 4. (A) Schematic representation of the interaction of the ERK/miR-21 signaling proteins by STRING analysis and PDCD4 was identified. (B) TCGA database analysis of 512 lung cancer patients showed a negative correlation between the expression of miR-21-5p and PDCD4. (C) PDCD4 was predicted as a target of miR-21-5p. The insert shows the miR-21-5p binds to the 3’UTR of PDCD4. Western blots showed a decreased expression of MEK1/2, STAT3, ERK1/2 and PDCD4 in miR-21-5p inhibitor (I) treated lung cancer cells and the overexpression of miR-21-5p (M) showed the reverse and bar graph of multiple experiments. (D) Real-time PCR analysis showed that CAF_R1 cells with silenced miR-21-5p expressed a significantly lower level of FAP, A-SAM, VIM and PDGFR. (E) CAF-associated cytokines, IL-6, IL-8 and HGF were also significantly reduced in the CAF_R1 cells with silenced miR-21-5p. (F) Tumor sphere formation assay. In both miR-21-5p inhibitor-treated P_AZDR1 and H1975+CAF_R1 cells, a significantly lower number of spheres was formed while mimic treatment reversed this phenomenon. **P<0.01; ***P<0.001. Scale bar: 100 μm.”
Q8: Figure 5F. Please provide quantitative analysis of the IHC data. This would both clarify and further support the authors’ conclusions.
A8. We sincerely thank the reviewer for this insightful suggestion. In the revised manuscript, we have provided quantitative analysis of the IHC data.
Please kindly see our revised result section.
“3.5. In vivo evaluation of therapeutic effects of Trametinib in combination with Osimertinib
….Finally, Immunohistochemical and quantitative analysis indicated that the combination treatment prominently reduced the expression of oncogenic markers including MEK, ERK, STAT3 and PDCD4, as well as increased E-cadherin. Notably, the expression of FAP was also prominently reduced by the combination treatment (Figure 5F).”
“Figure 5. Combined treatment of Trametinib and Osimertinib effectively inhibited the growth of P_AZDR1 (Osimertinib resistant) cells in vivo. (A) The flow chart for in vivo experimental design and treatment schedule. (B) Tumor curve over time shows that the combination of Trametinib and Osimertinib suppressed the tumor growth the most followed by the Trametinib alone group as compared to the vehicle and Osimertinib groups (no significant difference between these two groups). (C) Bodyweight over time curve demonstrated no apparent systematic toxicity in the mice receiving combined treatment. (D) Kaplan-Meier survival curve shows that the mice that received the combined treatment exhibited the highest survival ratio as compared to the rest. (E) Real-time PCR results of the tumor samples revealed that the miR-21-5p level was most significantly suppressed in the combined treatment followed by the Trametinib group while no significant difference was observed between vehicle and Osimertinib groups. (F) Immunostaining analysis of PDX tumor sections showed that the combined treatment most prominently suppressed the expression of MEK, ERK, STAT3 and PDCD4 but an increased E-cadherin was observed, comparing to other sections. *P<0.05; **P<0.01; ***P<0.001. NS, not significantly different. PDX: patient-derived xenograft.”

Reviewer 2 Report
The manuscript is dealing with a routinely found problem: the tki"s resistance. Especially of a new generation agent as osimertinib. The methodology of the research, the explanations/conclusions and the references are appropriate and solid enough to answer the question what therapeutical pathway we should consider in case of osimertinib failure. Maybe a paragraph about the probable use immunotherapy could give more details (line 475).
Author Response
Response to Reviewers:
Reviewer #2 (Comments to the Author):
Q1: The manuscript is dealing with a routinely found problem: the tki"s resistance. Especially of a new generation agent as osimertinib. The methodology of the research, the explanations/conclusions and the references are appropriate and solid enough to answer the question what therapeutical pathway we should consider in case of osimertinib failure. Maybe a paragraph about the probable use immunotherapy could give more details (line 475).
A1. We sincerely thank the reviewer for this insightful suggestion. In the revised manuscript, we added in the case of failure of osimertinib treatment. Maybe the passage about the possible use of immunotherapy, and about the probable use immunotherapy. Please kindly see our revised Disscussion section.
“4. Discussion
As one of the main challenges faced by clinicians treating non-small cell lung cancer (NSCLC), drug resistance continues to hinder efforts to slow the progression of the disease. Osimitinib is currently the standard first-line treatment for NSCLC patients with EGFR mutations. In addition, in about half of the first-line osimertinib failure cases, the molecular mechanism leading to resistance remains unknown. In view of osimertinib resistance, clinicians have explored other strategies, including reusing the first-generation EGFR TKI. In a recent study, osimertinib plus inter­mittent selumetinib (MEK1/MEK2 inhibitor) demonstrated preliminary antitumor activity in patients withEGFR-mutated NSCLC who progressed to prior EGFR TKI.The use of a combination of immune checkpoint inhibitors and tyrosine kinase inhibitors resulted in excessive toxicity without additional efficacy, while immunotherapy and chemotherapy, especially in combination with anti-angiogenic drugs, are more effective than those previously targeted It looks promising for the treated patients. MET and MEK inhibitors may overcome EGFR-independent resistance. Another acquired resistance mechanism of EGFR-TKI is the up-regulation of the RAS/RAF/MEK/ERK signaling pathway, which is the key to cell survival and proliferation; this may occur downstream of various other signaling pathways.In this report, we reveal the possible regulatory mechanism and inhibitory effect of the MEK inhibitor trametininb applied to MEK/ERK/miR-21 axis and PDCD4 in Osimertinib resistance.
We examined the potential underlying factors contributing to acquired resistance against Osimertinib. Comparatively, we found that clinical samples from Osimertinib-resistant patients contained a significantly higher level of ERK2 expression than the Osimertinib-sensitive counterparts (Figure 1C). This echoes the analysis from the database showing that a higher ERK2 expression was associated with worse survival in lung cancer patients (Figure 1D). We then disclosed the relationship between CAFs and Osimertinib resistance by co-culturing Osimertinib-sensitive NSCLC cells (P_AZDS1 and H1975) with CAF_R cells to increase cell viability under Osimertinib treatment (Figure 2D). This phenomenon was accompanied by the increase of specific marker expression and cytokine release in CAFs (Figure 2A, C), as well as the increase of tumor sphere formation and up-regulation of MEK1/2, ERK1/2, STAT3 plus β-catenin in NSCLC cells (Figure 2E, F). These findings highlighted the critical roles of CAFs and MEK/ERK signaling in Osimertinib resistance. Regarding the roles of CAFs in drug resistance, it has been shown that CAFs secreted IL-6 to enhance epithelial-mesenchymal transition (EMT) and Cisplatin resistance in NSCLC [28]. Though generally considered critical for metastasis, the role of EMT in drug resistance had been described [29]. In addition to promoting EMT, some studies also suggested that the drug-scavenging ability of CAFs may contribute to Gemcitabine failure in pancreatic cancer, while other studies demonstrated that CAFs can inhibit reactive oxygen species (ROS) production to antagonize chemotherapy-induced cell death in prostate cancer [30,31]. Concerning therapeutic strategies aiming at CAFs, it has been reported that the application of inhibitors for STAT3 and MEK1/2 could suppress tumorigenesis in the organotypic model of esophageal cancer [32]. Thus, it is rational to try MEK inhibitors to overcome Osimertinib resistance from the abovementioned results.”
“A recent report from Gong et al demonstrated that activated EGFR signaling results in a rapid inhibition of TNF mRNA through miR-21 induction, and inhibition of EGFR results in amplified TNF mRNA and reduced miR-21 expression levels in lung cancer They also showed TNF activated NF-kB, which in turn induces the transcription of TNF mRNA in a feedforward control [42]. In fact, it has been reported earlier that aberrantly increased expression of miR-21 was observed in lung carcinogenesis in smokers and non-smokers, and this is further enhanced by activation of EGFR signaling pathway [43]. In our current study, we found that CAF_R1 cells had higher expression of miR-21 (Figure 2B), and it was effectively suppressed by the MEK inhibitor Trametinib (Figure 3C). These were compatible with previous reports because the CAF_R1 cells were derived from P_AZDR1 cells, of which the EGFR signaling was theoretically activated in large amount due to their characteristics of drug resistance. Regarding the cross-talk between NF-kB and STAT3 signaling, it has been shown in a variety of cancers. NF-kB and STAT3 can bind the same promoters/enhancers and share many downstream genes [44]. The NF-kB/IL-6/STAT3 axis has also been well established [45,46]. STAT3 can also activate NF-kB, but the reported mechanism is still relatively less. Here we identified that the combination treatment prominently reduced the expression of STAT3, which further resulted in the reduced secretion of IL-6. These events might switch off the activation of NF-kB and TNF mRNA feedforward loop and sensitize the NSCLC cells towards the treatment. Combination treatments have been suggested to provide solutions for NSCLC patients encountering resistance after the administration of target therapies [47-49]. The combinations were not limited to target therapy or small molecule therapy but also included chemotherapy and immunotherapy. When focusing on the combination of Osimertinib and MEK inhibitor, there have been some reports addressing this issue. It was reported that this Osimertinib plus Trametinib enhanced induction of apoptosis in EGFR-mutant NSCLC cells but not in EGFR wild-type NSCLC cells, and was very effective in killing cell clones with primary intrinsic resistance to Osimertinib [50]. The same group also reported that the combination of a MEK or ERK inhibitor with a first-generation or second-generation EGFR-TKI also very effectively inhibited the growth of Osimertinib-resistant cells, although these cell lines were cross-resistant to first-generation or second-generation EGFR-TKIs [51]. Our study essentially verified the effectiveness of such a combination, but we also provided evidence of CAF involvement in this process and disclosed the underlying mechanisms focusing on miR-21/PDCD4.”
